# Association of *VDR* Polymorphisms (*FokI*, *ApaI*, and *TaqI*) with Susceptibility to Lumbar Disc Herniation: Systematic Review, Meta-Analysis, Trial Sequential Analysis, and Transcriptional Prediction

**DOI:** 10.3390/medicina61050882

**Published:** 2025-05-12

**Authors:** Alireza Sheikhi, Mohsen Nabiuni, Soha Zia, Masoud Sadeghi, Annette B. Brühl, Serge Brand

**Affiliations:** 1Firoozgar Clinical Research Development Center (FCRDC), Iran University of Medical Sciences (IUMS), Tehran 1593748711, Iran; alirezasheykhi771@gmail.com (A.S.); nabiuni_m@yahoo.com (M.N.); 2Department of Pathology, School of Medicine, Tehran University of Medical Sciences, Tehran 1416634793, Iran; dr.sohazia@yahoo.com; 3Medical Biology Research Center, Health Technology Institute, Kermanshah University of Medical Sciences, Kermanshah 671551616, Iran; sadeghi_mbrc@yahoo.com; 4Center for Affective, Stress and Sleep Disorders, Psychiatric Clinics, University of Basel, 4002 Basel, Switzerland; annette.bruehl@upk.ch; 5Sleep Disorders Research Center, Kermanshah University of Medical Sciences, Kermanshah 6714869914, Iran; 6Substance Abuse Prevention Research Center, Kermanshah University of Medical Sciences, Kermanshah 6714869914, Iran; 7Division of Sport Science and Psychosocial Health, Department of Sport, Exercise and Health, University of Basel, 4031 Basel, Switzerland; 8School of Medicine, Tehran University of Medical Sciences, Tehran 1339973111, Iran; 9Center for Disaster Psychiatry and Disaster Psychology, Center of Competence of Disaster Medicine of the Swiss Armed Forces, Psychiatric Clinics, University of Basel, 4002 Basel, Switzerland

**Keywords:** lumbar disc herniation, vitamin D receptor, genetic polymorphism, meta-analysis, allelic variation

## Abstract

*Background and Objectives*: Lumbar disc herniation (LDH) is influenced by genetic, mechanical, and behavioral factors, with genetic predisposition playing a key role. *Vitamin D receptor* (*VDR*) polymorphisms have been implicated in LDH susceptibility, warranting further investigation. This study aimed to assess the association between *VDR* polymorphisms (*FokI*, *ApaI*, and *TaqI*) and LDH risk through a systematic review, meta-analysis, and trial sequential analysis (TSA). *Materials and Methods*: A systematic literature search was conducted across PubMed, Web of Science, Scopus, Cochrane Library, and CNKI, up until 30 January 2025. A meta-analysis was performed using Review Manager 5.3, with odds ratios (ORs) and 95% confidence intervals (CIs), and heterogeneity assessed via the I^2^ statistic. The publication bias and TSA were evaluated using CMA 3.0 and TSA software to ensure the reliability of the results. The FATHMM-XF method was applied to predict the functional effect of coding and non-coding polymorphisms. *Results*: From 79 records, 10 studies were entered into the meta-analysis. The meta-analysis results showed no significant association of *FokI* and *ApaI* polymorphisms with LDH, while *TaqI* exhibited a protective effect, particularly in Asian populations and larger studies. The subgroup analysis revealed significant ethnicity-specific associations for *TaqI*, with stronger effects observed in Asian compared to Caucasian individuals. The trial sequential analysis indicated that additional studies are required to confirm the findings for *FokI*, while the recessive model of *TaqI* polymorphism showed a near-sufficient sample size for reliable conclusions. *Conclusions*: The *TaqI* polymorphism, particularly the tt genotype, appears to have a protective effect against LDH, especially in Asian populations and larger studies. However, further large-scale, multi-ethnic research is needed to confirm these findings and explore underlying biological mechanisms.

## 1. Introduction

Lumbar disc herniation (LDH) occurs when the fibrous annulus of the intervertebral disc ruptures, leading to the extrusion of the nucleus pulposus. This extrusion places pressure on the spinal nerve root, causing radiculopathy [1] and, in some cases, affecting the cauda equina [2], with a prevalence of 2% in LDH cases [3]. The development of LDH is influenced by genetic, mechanical, and behavioral factors [4]. The prevalence of symptomatic LDH is estimated to be less than 5% of the population [5,6], while in Mexico, it has been reported at over 23.12% [7]. In the United States, LDH ranks as the second leading cause of work-related disability [8].

A body mass index exceeding 30 kg/m^2^, a family history of LDH, poor sitting posture, prolonged daily sitting (over six hours), and a history of lower back trauma are key risk factors for adolescents and young adults with LDH [9]. Research indicates that a significant proportion of LDH cases have a genetic basis [10,11,12], suggesting that genetic factors may play a more dominant role than environmental influences in the onset and progression of LDH [13]. The genetic susceptibility to LDH likely arises from a complex interaction of mutations affecting the nucleus pulposus, annulus fibrosus, and cartilaginous end plates [11].

Vitamin D plays a crucial role in regulating immune responses and influences key cell populations involved in disc herniation by suppressing neurotoxic agents and acting on neurotrophins [14]. Low serum 25(OH)D levels are significantly associated with LDH, and even in tropical climates, routine screening for vitamin D deficiency and supplementation should be considered essential for LDH patients [15]. Additionally, polymorphisms in the *vitamin D receptor* (*VDR*) gene contribute to the development of LDH, highlighting a genetic link to the condition [15,16,17,18,19].

This study aimed to systematically evaluate the association between *VDR* polymorphisms (*FokI*, *ApaI*, and *TaqI*) and susceptibility to LDH. Given the significant role of genetic factors in LDH development, we will conduct a comprehensive systematic review, meta-analysis, and trial sequential analysis to determine whether these specific *VDR* gene variants contribute to an increased risk of LDH. By synthesizing the existing evidence, this research seeks to clarify the genetic predisposition to LDH and its potential implications for risk assessment and targeted interventions.

## 2. Materials and Methods

The meta-analysis was managed following the protocols of the Preferred Reporting Items for Systematic Reviews and Meta-Analyses (PRISMA) [20]. The PECO framework was as follows: Population (P): Individuals with LDH. Exposure (E): Presence of *VDR* polymorphisms (*FokI*, *ApaI*, and *TaqI*). Comparator (C): Individuals without LDH (healthy controls). Outcome (O): Susceptibility to LDH. The study was not registered in any database.

### 2.1. Literature Search

Five electronic databases, PubMed, Web of Science, Scopus, Cochrane Library, and CNKI, were searched up to 30 January 2025, without any restrictions. The search terms were (“lumbar disc herniation” or “disc herniation” or “herniated disk” or “ruptured disk” or “slipped disc”) and (“*VDR*” or “*FokI*” or “*ApaI*” or “*TaqI*” or “*vitamin D receptor*”) and (“polymorphism*” or “variant*” or “genotype*” or “allele*”). We also examined the references of eligible studies and conducted a manual review of articles to identify potentially relevant publications, in addition to searching electronic sources like Google Scholar. Notably, ethical approval was not required, as this study involved retrieving and synthesizing data from previously published articles.

### 2.2. Eligibility Criteria

The criteria for inclusion encompass case–control studies that have a case group with LDH or systemic diseases and a control group without LDH or systemic diseases. In addition, studies reported distributions of genotypes of *VDR* polymorphisms in both groups. The criteria for exclusion consist of studies that involve participants with a history or diagnosis of any systemic diseases that overlap with LDH, reviews, meta-analyses, studies with incomplete data, and studies with duplicate data.

### 2.3. Data Extraction

To ensure consistency in the screening criteria and data collection process, two authors independently reviewed the literature and extracted the data.

### 2.4. Statistical Analysis and Data Synthesis

We used Review Manager, version 5.3, for the meta-analysis and presented the data as odds ratios (ORs) and 95% confidence intervals (CIs). The studies’ heterogeneity was evaluated using the I^2^ statistic, with the significance level set at *p* < 0.05 [17,18], ranging from 0 to 100%. We used of random-effects model, if heterogeneity was more than 50% (*P*_heterogeneity_ < 0.10) [21]; otherwise, a fixed-effect model was used [22]. Publication bias was assessed using a funnel plot and Begg’s and Egger’s tests, with the significance level set at *p* < 0.10 [23,24]. Comprehensive Meta-Analysis version 3.0 (CMA 3.0) software was employed for assessing publication bias, conducting meta-regression analysis, and performing sensitivity analyses.

A trial sequential analysis (TSA) was conducted using TSA software (version 0.9.5.10 beta) [25]. The required information size (RIS) for blood cytokine levels was determined with an alpha risk of 5% and a beta risk of 20%. If the Z-curve intersected the RIS, it suggested that the studies included a sufficient number of cases, making the conclusion reliable.

### 2.5. Transcriptional Prediction Method

The FATHMM-XF method (https://fathmm.biocompute.org.uk/fathmm-xf/, accessed on 5 January 2025) was applied to predict the functional effect of coding or non-coding polymorphisms. FATHMM-XF results were divided into two categories, as follows: pathogenic (*p* > 0.50) and neutral or benign (*p* < 0.50) [26,27].

## 3. Results

### 3.1. Study Selection

Figure 1 outlines the study selection process for the systematic review and meta-analysis. Initially, 79 records were identified from various databases, with no additional records from other sources. After removing the duplicates, 33 records remained for screening, of which 19 were excluded (conference papers, records reporting *VDR* polymorphisms in lumbar disc degeneration, and records without any information on *VDR* polymorphisms). Fourteen full-text articles were assessed for eligibility, but four were excluded due to patient overlap, spinal trauma cases, or duplicate data. Ultimately, 10 articles/studies [15,16,17,18,19,28,29,30,31,32] were included in both the systematic review and meta-analysis.

### 3.2. Characteristics of the Studies

Table 1 presents the basic characteristics of the included studies in the meta-analysis. The studies span from 2014 to 2024 and cover multiple countries, including Italy, Bulgaria, Turkey, China, Sri Lanka, and others. The majority of the studies focus on Caucasian populations (from Italy, Bulgaria, and Turkey), while three studies include Asian populations (from China and Sri Lanka). The genotyping methods used include Polymerase Chain Reaction-Restriction Fragment Length Polymorphism (PCR-RFLP), PCR, and real-time PCR with TaqMan, with PCR-RFLP being the most commonly used technique. Notably, one study [31] did not report the genotyping method. The *VDR* polymorphisms analyzed include *FokI*, *ApaI*, and *TaqI*. The Newcastle–Ottawa Scale (NOS) scores, which assess study quality, ranged from 7 to 9, indicating generally high-quality research.

### 3.3. Transcriptional Prediction

The FATHMM-XF predictions for polymorphisms on Chromosome 12 indicate that all three variants analyzed are benign. The *rs2228570* (*FokI*) polymorphism has a moderate coding score of 0.433632, suggesting a low likelihood of functional impact. The *rs7975232* (*ApaI*) polymorphism has a very low coding score of 0.053047, classifying it as benign with little to no effect on gene function. The third variant (*rs731236* (*TaqI*)) has an extremely low coding score of 0.005901, meaning it is highly likely to be benign, with high confidence. Overall, these results suggest that none of the analyzed polymorphisms are likely to cause significant functional changes in the VDR gene or its associated protein (Table 2).

### 3.4. Genotype Distribution

Table 3 presents the genotype distributions of the *VDR* polymorphisms (*FokI*, *ApaI*, and *TaqI*) across multiple studies, comparing cases and controls while reporting Hardy–Weinberg Equilibrium (HWE) *p*-values for control groups. For *FokI* (*rs2228570*), most studies show equilibrium (*p* > 0.05), except Gaydarski, 2023 [28] (*p* = 0.006), indicating potential deviation. *ApaI* (*rs7975232*) is analyzed in three studies, with all maintaining HWE, suggesting a balanced control population. *TaqI* (*rs731236*) shows some variation, with Withanage, 2018 [15] (*p* = 0.027) slightly deviating from equilibrium.

### 3.5. Pooled Results

Table 4 summarizes the pooled results of forest plots (see Appendix A) for the association of *VDR* polymorphisms with LDH risk. For *FokI*, all models yield non-significant results (*p* > 0.05), with moderate-to-large heterogeneity (I^2^ = 39–69%), suggesting variability among the studies. The *ApaI* polymorphism shows no significant association with the LDH risk in any genetic model (*p* > 0.05), and low heterogeneity (I^2^ = 0–37%), indicating consistent results across the studies. Conversely, *TaqI* exhibits statistically significant associations in the homozygous model (tt vs. TT: OR = 0.55, *p* = 0.0006) and the recessive model (tt vs. TT + Tt: OR = 0.59, *p* = 0.001), implying a protective role of the tt genotype. Overall, *TaqI* appears to be the most relevant polymorphism, warranting further investigation, while *FokI* and *ApaI* show no clear associations with disease risk.

### 3.6. Subgroup Analysis

The subgroup analysis of the *VDR* polymorphisms (*FokI* and *TaqI*) by ethnicity and sample size provides further insights into the potential genetic associations (Table 5). For *FokI*, no significant associations were found across the ethnic groups or sample sizes (*p* > 0.05), and heterogeneity remained high in several comparisons (I^2^ up to 83%), indicating study variability. *TaqI*, however, showed stronger associations in Asian individuals, particularly in the allelic (t vs. T: OR = 0.69, *p* = 0.0005), homozygous (tt vs. TT: OR = 0.44, *p* = 0.003), and recessive (tt + Tt vs. TT: OR = 0.64, *p* = 0.001) models, all with low heterogeneity (I^2^ ≤ 48%). In contrast, Caucasian individuals showed weaker or non-significant associations, except in the recessive model (tt vs. TT + Tt: OR = 0.64, *p* = 0.03). When stratified by sample size, significant associations were evident for *TaqI* in the larger studies (≥200 participants), particularly in the allelic (OR = 0.73, *p* = 0.0001) and homozygous (OR = 0.48, *p* = 0.0001) models, reinforcing the robustness of these findings. However, smaller studies (<200 participants) lacked significance, often displaying high heterogeneity (I^2^ up to 94%), which suggests potential sample bias or study differences. These findings highlight the ethnicity-specific effects of *TaqI*, warranting further investigation in larger, well-powered studies.

### 3.7. Meta-Regression Analysis

The meta-regression analysis for *FokI* and *TaqI* polymorphisms assessed the impact of publication year and sample size on the genetic associations (Table 6). None of the models for *FokI* showed significant effects of publication year (*p* ≥ 0.48) or sample size (*p* ≥ 0.54), suggesting that temporal trends and study sizes did not influence the observed genetic associations. Similarly, for *TaqI*, neither the publication year (*p* ≥ 0.75) nor sample size (*p* ≥ 0.56) showed a significant effect, except for the homozygous model (tt vs. TT), where the sample size had a borderline effect (*p* = 0.063), indicating a possible influence of study scale on this association. The lack of significant meta-regression findings suggests that the heterogeneity in the pooled results is likely due to other factors, such as ethnic differences or methodological variations rather than the study size or publication bias.

### 3.8. Sensitivity Analysis

The one-study-removed and cumulative analyses confirmed the robustness of the initial pooled results, indicating that no single study significantly influenced the overall findings. This consistency suggests that the association between *VDR* polymorphisms and LDH remains stable and reliable, regardless of the sequential removal of individual studies or the progressive accumulation of data. We removed one study with a deviation from HWE [28] for the *VDR FokI* (*rs2228570*) polymorphism, in which just heterogeneity reduced to 38% in the allelic model, to 39% in the homozygous model, to 48% in the heterozygous model, to 49% in the dominant model, and to 0% in the recessive model. However, there was no association based on all of the models in line with the initial analyses.

### 3.9. TSA

Appendix A shows the TSAs for five genetic models for the *FokI* and *TaqI* polymorphisms. For five genetic models, the declining trend of the Z-curve implies that additional studies may shift the overall evidence, highlighting the need for larger, well-powered investigations to clarify the genetic influence of the *FokI* polymorphism on the LDH risk. The total sample sizes of 3994 and 916 patients remain below the RIS (6187 and 1095) for the allelic and homozygous models of the *TaqI* polymorphism, respectively, indicating insufficient statistical power for a conclusive finding. The unstable trend of the Z-curve implies that further research with larger sample sizes is necessary to determine the precise role of the *TaqI* polymorphism in LDH susceptibility for both allelic and homozygous models. The sample size of 1784 and 1997 patients is far below the RIS (15,040 and 7538) for heterozygous and dominant models of the *TaqI* polymorphism, respectively, suggesting that the current evidence is insufficient to establish a definitive conclusion. The wide futility boundary further indicates that more studies with larger sample sizes are needed to determine whether the *TaqI* polymorphism significantly influences the LDH risk. However, in the recessive model, the Z-curve trends upward and crosses the significance boundary (red lines), suggesting a statistically significant association. The sample size of 1363 patients is very close to the RIS of 1409, indicating that the available evidence is nearly sufficient to draw a reliable conclusion. Since the Z-curve crosses the futility boundary, it suggests that further studies may not significantly change the overall findings.

### 3.10. Publication Bias

Appendix A shows the funnel plots for five genetic models for *FokI*, *ApaI*, and *TaqI* polymorphisms. There was significant publication bias only for the *TaqI* polymorphisms in the recessive model (*p*-values of Egger’s test: 0.0596 and Begg’s test: 0.0500).

## 4. Discussion

This systematic review and meta-analysis suggest that the *TaqI* polymorphism may have a protective role in LDH, especially in Asian individuals, while *FokI* and *ApaI* show no significant associations. The insufficient statistical power, high heterogeneity, and publication bias highlight the need for larger, ethnically diverse studies to confirm these findings.

A 2021 meta-analysis [33] reported a weak association between the *TaqI* (*rs731236*) polymorphism and susceptibility to herniated nucleus pulposus in both genotype and allele distributions. Another meta-analysis from the same year [34] found evidence linking the *TaqI* polymorphism of the VDR gene to an increased risk of developing lumbar spine pathologies, including LDH. However, our meta-analysis identifies a protective role of the *TaqI* polymorphism, particularly for the “tt” genotype. A key reason for the differences among these three meta-analyses is that the previous meta-analyses did not specifically select LDH cases but included a mix of lumbar spine pathologies. In contrast, our analysis focused exclusively on LDH cases, excluding lumbar disc degeneration and other spinal disorders.

In the present meta-analysis, *FokI* and *ApaI* polymorphisms showed no significant association with LDH risk. In contrast, the *TaqI* polymorphism demonstrated a significant protective effect in the homozygous and recessive models, with stronger associations observed in Asian populations and larger studies. The differences in the association between *VDR* polymorphisms (*FokI*, *ApaI*, and *TaqI*) and LDH risk could be attributed to several factors. First, different *VDR* gene polymorphisms influence the function of *VDR* in distinct ways [35,36]. The *TaqI* polymorphism may play a more significant role in regulating vitamin D-related pathways involved in disc metabolism and inflammation, while *FokI* and *ApaI* might have less direct effects on these processes. Second, the observed stronger association of the *TaqI* polymorphism in Asian populations suggests that the genetic background plays a crucial role. Variations in allele frequencies across different ethnic groups may lead to population-specific effects, with some polymorphisms exerting a greater influence on the LDH susceptibility in certain populations. Third, the moderate-to-large heterogeneity observed in the *FokI* studies suggests that variability in study methodologies, sample sizes, and population characteristics may have influenced the results. In contrast, the low heterogeneity for *ApaI* indicates more consistent findings across studies, though it did not show a significant association with the LDH risk. Fourth, the *TaqI* polymorphism is located in the 3′ untranslated region (3′ UTR) of the VDR gene [37,38], which may affect mRNA stability and gene expression [38], potentially influencing the LDH risk. On the other hand, *FokI* and *ApaI* are situated in coding or intronic regions [37,38,39], which may have less direct functional consequences on VDR activity in intervertebral disc health.

### Limitations

Potential Deviation from HWE: some studies showed deviations from HWE, which may indicate issues with population representativeness or genotyping errors, potentially affecting the reliability of the findings.High Heterogeneity Among Studies: the *FokI* polymorphism exhibited moderate-to-large heterogeneity (I^2^ = 39–69%), and subgroup analyses showed heterogeneity up to 83% for certain comparisons, suggesting variability in study methodologies, sample populations, or environmental factors that may influence the pooled results.Insufficient Statistical Power for Some Models: the TSA indicated that the total sample sizes for several genetic models, particularly for *FokI* and *TaqI* (allelic, homozygous, heterozygous, and dominant models), remained below the RIS, implying that additional well-powered studies are needed to reach conclusive findings.Potential Publication Bias: the funnel plot analysis revealed significant publication bias for the *TaqI* polymorphism in the recessive model, suggesting that smaller studies with null results may be underrepresented, potentially skewing the overall conclusions.

## 5. Conclusions

The FATHMM-XF predictions classify *FokI* (*rs2228570*), *ApaI* (*rs7975232*), and *TaqI* (*rs731236*) polymorphisms as benign, suggesting minimal functional impact on the *VDR* gene. These bioinformatics predictions align with the meta-analysis, which found no significant association between *FokI* and *ApaI* polymorphisms and LDH risk, though *FokI* exhibited moderate-to-large heterogeneity, while *ApaI* showed low heterogeneity. However, *TaqI* demonstrated a significant protective effect in homozygous and recessive models, particularly in Asian populations and larger studies (≥200 participants), though the publication bias in the recessive model necessitates cautious interpretation. Additionally, the TSA analysis indicates that the current sample sizes remain below the RIS in several models, highlighting the need for further large-scale studies to confirm these associations. Overall, while the computational predictions classify these polymorphisms as benign, the meta-analysis suggests a possible protective role for *TaqI*, emphasizing the necessity of additional research to validate these findings.

While genetic studies show that the *TaqI* (*rs731236*) polymorphism has a protective effect in homozygous and recessive models; FATHMM-XF predicts it as benign based on computational analysis. The protective effect is observed in real-world populations, suggesting a statistical association with lower disease risk, but this does not necessarily mean the polymorphism has a strong functional impact on the *VDR* gene itself. Computational tools like FATHMM-XF primarily evaluate sequence conservation and potential structural alterations, which may overlook regulatory or population-specific effects. Additionally, the influence of the polymorphism might be context-dependent, meaning its impact varies depending on environmental or physiological factors.

### 5.1. Clinical Significance

The findings suggest that the *TaqI* polymorphism, particularly the tt genotype, may confer protection against LDH, which could have implications for genetic risk assessment and personalized treatment approaches. Genetic screening for *TaqI* polymorphism may help identify individuals at lower or higher risk of developing LDH, particularly in Asian populations. However, *FokI* and *ApaI* do not appear to be clinically relevant genetic markers, limiting their utility in risk prediction models. Given the remaining uncertainties due to insufficient sample sizes, these findings should be interpreted with caution until validated by larger, well-powered studies.

### 5.2. Future Directions

Future research should focus on large-scale, multi-ethnic studies with standardized methodologies to further clarify the role of the *TaqI* polymorphism in LDH susceptibility. Functional studies exploring the biological mechanisms behind the protective effect of the tt genotype could provide deeper insights into the role of *VDR* gene variations in spinal degeneration. Additionally, gene–environment interactions, such as the impact of vitamin D levels, lifestyle factors, and occupational strain, should be investigated to determine their influence on LDH risk. Lastly, meta-analyses incorporating individual patient data may help reduce heterogeneity and improve the accuracy of genetic associations.

## Figures and Tables

**Figure 1 medicina-61-00882-f001:**
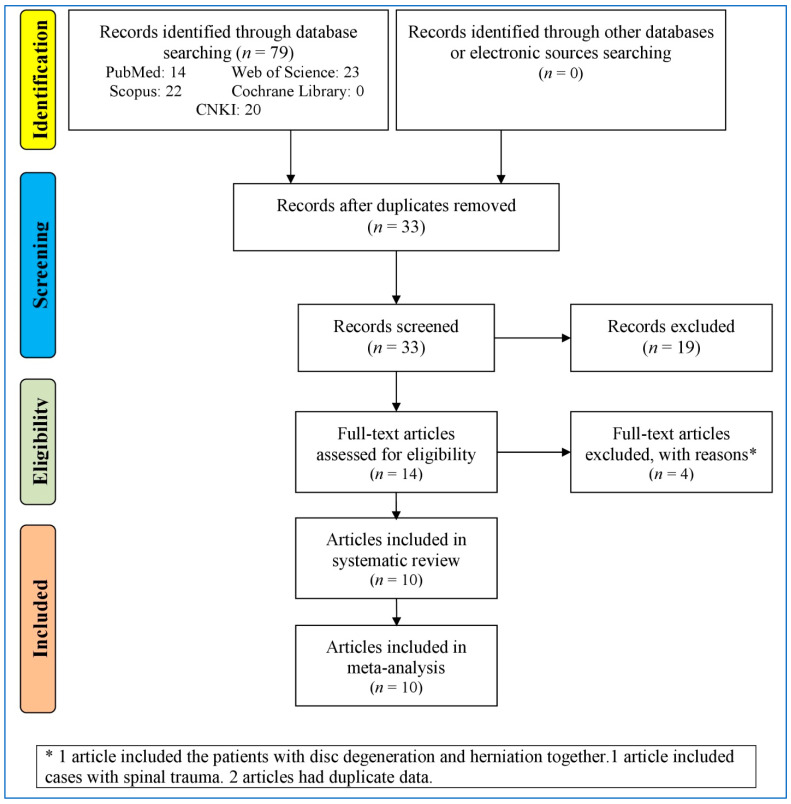
Flowchart of the study selection.

**Table 1 medicina-61-00882-t001:** Basic characteristics of the studies.

First Author, Publication Year	Country	Ethnicity	Genotyping Method	Polymorphism of VDR	NOS Score
Colombini, 2014 [18]	Italy	Caucasian	PCR-RFLP	*FokI*	8
Colombini, 2015 [16]	Italy	Caucasian	PCR-RFLP	*FokI*	8
Colombini, 2016 [17]	Italy	Caucasian	PCR-RFLP	*ApaI and TaqI*	8
Gaydarski, 2023 [28]	Bulgaria	Caucasian	PCR-RFLP	*FokI*	9
Hekimoğlu, 2020 [29]	Turkey	Caucasian	PCR-RFLP	*FokI and TaqI*	9
Li, 2018 [30]	China	Asian	PCR	*FokI, ApaI, and TaqI*	9
Sansoni, 2016 [31]	Italy	Caucasian	NR	*FokI*	8
Serifoğlu, 2024 [32]	Turkey	Caucasian	PCR	*TaqI*	9
Withanage, 2018 [15]	Sri Lanka	Asian	PCR	*FokI and TaqI*	9
Yang, 2019 [19]	China	Asian	Real-time PCR and TaqMan	*FokI, ApaI, and TaqI*	7

VDR: Vitamin D receptor. NR: not reported. PCR: polymerase chain reaction. RFLP: restriction fragment length polymorphism.

**Table 2 medicina-61-00882-t002:** FATHMM-XF predictions for polymorphisms on Chromosome 12.

Polymorphism	Chromosome	Position	Variant	Coding Score	Non-Coding Score	Further Information
*rs2228570* (*FokI*)	12	47879112	A/G	0.433632	-	Benign
*rs7975232* (*ApaI*)	12	47845054	C/A	-	0.053047	Benign
*rs731236* (*TaqI*)	12	47844974	A/G	0.005901	-	Benign (high confidence)

**Table 3 medicina-61-00882-t003:** Prevalence of genotypes of VDR polymorphisms.

**Polymorphism**	**First Author, Publication Year**	**No. Cases/Controls**	**Case**			**Control**			***p*-Value of HWE for Control Group**
***VDR FokI*** (***rs2228570***)			**FF**	**Ff**	**ff**	**FF**	**Ff**	**ff**	
	Colombini, 2014 [18]	89/220	37	40	12	89	99	32	0.601
	Colombini, 2015 [16]	48/127	25	19	4	51	56	20	0.483
	Gaydarski, 2023 [28]	60/60	12	40	8	29	31	0	0.006
	Hekimoğlu, 2020 [29]	72/81	37	34	1	55	24	2	0.744
	Li, 2018 [30]	120/120	44	53	23	31	66	23	0.250
	Sansoni, 2016 [31]	110/110	53	44	13	46	54	10	0.296
	Withanage, 2018 [15]	51/68	34	16	1	38	26	4	0.871
	Yang, 2019 [19]	266/485	67	123	76	134	225	126	0.829
***VDR ApaI*** (***rs7975232***)			AA	Aa	aa	AA	Aa	aa	
	Colombini, 2016 [17]	88/252	26	51	11	92	108	52	0.055
	Li, 2018 [30]	120/120	13	47	60	16	48	56	0.272
	Yang, 2019 [19]	266/485	130	116	20	244	191	50	0.169
***VDR**TaqI*** (***rs731236***)			TT	Tt	tt	TT	Tt	tt	
	Colombini, 2016 [17]	88/252	37	40	11	106	109	37	0.303
	Hekimoğlu, 2020 [29]	72/81	19	45	8	39	30	12	0.133
	Li, 2018 [30]	120/120	114	6	0	109	11	0	0.598
	Serifoğlu, 2024 [32]	248/146	107	113	28	45	72	29	0.983
	Withanage, 2018 [15]	51/68	31	16	4	25	39	4	0.027
	Yang, 2019 [19]	266/485	118	131	17	176	246	63	0.109

HWE: Hardy–Weinberg Equilibrium.

**Table 4 medicina-61-00882-t004:** Summary of pooled results of the forest plots.

Polymorphism	Genetic Model	OR	95%CI		Z	*p*-Value	I^2^	*P_heterogeneity_*
			**Min.**	**Max.**				
** *FokI* **	f vs. F	1.03	0.80	1.32	0.19	0.85	66%	0.004
	ff vs. FF	1.04	0.79	1.38	0.30	0.77	39%	0.120
	Ff vs. FF	1.01	0.71	1.43	0.05	0.96	64%	0.007
	ff + Ff vs. FF	1.02	0.71	1.46	0.10	0.92	69%	0.002
	ff vs. FF + Ff	1.08	0.85	1.38	0.64	0.52	11%	0.340
ApaI	a vs. A	1.00	0.84	1.19	0.04	0.97	0%	0.750
	aa vs. AA	0.86	0.58	1.27	0.78	0.44	0%	0.500
	Aa vs. AA	1.25	0.97	1.62	1.70	0.09	0%	0.490
	aa + Aa vs. AA	1.14	0.89	1.46	1.06	0.29	0%	0.680
	aa vs. Aa + AA	0.81	0.59	1.11	1.30	0.19	37%	0.200
** *TaqI* **	t vs. T	0.80	0.62	1.02	1.77	0.08	58%	0.04
	tt vs. TT	0.55	0.39	0.77	3.45	**0.0006**	37%	0.17
	Tt vs. TT	0.84	0.52	1.34	0.75	0.46	76%	0.0009
	tt + Tt vs. TT	0.79	0.51	1.21	1.08	0.28	74%	0.002
	tt vs. TT + Tt	0.59	0.43	0.81	3.28	**0.001**	0%	0.50

OR: odds ratio. CI: confidence interval. The bold number means statistically significant datum (*p* < 0.05).

**Table 5 medicina-61-00882-t005:** Subgroup analysis.

Polymorphism	Genetic Model	Variable	Subgroup (N)	OR	95%CI		*p*-Value	I^2^
					**Min.**	**Max.**		
** *FokI* **	f vs. F	Ethnicity	Caucasian (5)	1.15	0.76	1.75	0.50	75%
			Asian (3)	0.91	0.67	1.23	0.53	51%
		Sample size	≥200 (4)	0.99	0.85	1.15	0.92	0%
			<200 (4)	1.14	0.58	2.24	0.70	83%
	ff vs. FF	Ethnicity	Caucasian (5)	1.03	0.43	2.45	0.95	55%
			Asian (3)	1.02	0.72	1.45	0.92	31%
		Sample size	≥200 (4)	1.04	0.77	1.41	0.79	0%
			<200 (4)	1.06	0.16	7.06	0.95	70%
	Ff vs. FF	Ethnicity	Caucasian (5)	1.21	0.70	2.08	0.49	72%
			Asian (3)	0.87	0.66	1.16	0.35	49%
		Sample size	≥200 (4)	0.88	0.69	1.12	0.29	30%
			<200 (4)	1.32	0.63	2.77	0.46	75%
	ff + Ff vs. FF	Ethnicity	Caucasian (5)	1.23	0.70	2.17	0.48	76%
			Asian (3)	0.81	0.51	1.29	0.37	57%
		Sample size	≥200 (4)	0.92	0.73	1.15	0.45	28%
			<200 (4)	1.29	0.56	3.00	0.55	82%
	ff vs. FF + Ff	Ethnicity	Caucasian (5)	1.10	0.71	1.71	0.68	39%
			Asian (3)	1.08	0.80	1.44	0.62	0%
		Sample size	≥200 (4)	1.10	0.85	1.43	0.48	0%
			<200 (4)	0.88	0.19	4.14	0.87	56%
** *TaqI* **	t vs. T	Ethnicity	Caucasian (3)	0.94	0.60	1.48	0.79	78%
			Asian (3)	0.69	0.57	0.85	**0.0005**	0%
		Sample size	≥200 (4)	0.73	0.63	0.86	**0.0001**	6%
			<200 (2)	0.94	0.38	2.33	0.90	83%
	tt vs. TT	Ethnicity	Caucasian (3)	0.71	0.36	1.43	0.34	56%
			Asian (3)	0.44	0.26	0.75	**0.003**	0%
		Sample size	≥200 (4)	0.48	0.33	0.70	**0.0001**	27%
			<200 (2)	1.14	0.49	2.70	0.76	0%
	Tt vs. TT	Ethnicity	Caucasian (3)	1.24	0.55	2.76	0.61	84%
			Asian (3)	0.57	0.32	1.01	0.05	55%
		Sample size	≥200 (4)	0.78	0.62	0.98	**0.03**	0%
			<200 (2)	1.02	0.11	9.04	0.99	94%
	tt + Tt vs. TT	Ethnicity	Caucasian (3)	1.11	0.51	2.41	0.80	85%
			Asian (3)	0.64	0.49	0.84	**0.001**	24%
		Sample size	≥200 (4)	0.71	0.58	0.89	**0.002**	0%
			<200 (2)	0.99	0.15	6.59	0.99	93%
	tt vs. TT + Tt	Ethnicity	Caucasian (3)	0.64	0.42	0.96	**0.03**	0%
			Asian (3)	0.52	0.31	0.87	**0.01**	48%
		Sample size	≥200 (4)	0.55	0.39	0.77	**0.0007**	0%
			<200 (2)	0.87	0.40	1.93	0.74	0%

OR: odds ratio. CI: confidence interval. N: number of studies. The bold number means statistically significant datum (*p* < 0.05).

**Table 6 medicina-61-00882-t006:** Meta-regression analysis.

Polymorphism	Genetic Model	Variable	Coefficient	95%CI		Z-Value	*p*-Value
				**Min.**	**Max.**		
** *FokI* **	f vs. F	Publication year	<0.0001	−0.0002	0.0003	0.14	0.8848
		Sample size	−0.0000	−0.0014	0.0014	−0.05	0.9634
	ff vs. FF	Publication year	−0.0002	−0.0006	0.0003	−0.69	0.4881
		Sample size	0.0007	−0.0015	0.0028	0.61	0.5444
	Ff vs. FF	Publication year	<0.0001	−0.0003	0.0004	0.17	0.8626
		Sample size	−0.0002	−0.0021	0.0018	−0.17	0.8688
	ff + Ff vs. FF	Publication year	<0.0001	−0.0003	0.0004	0.16	0.8726
		Sample size	−0.0001	−0.0021	0.0019	−0.11	0.9104
	ff vs. FF + Ff	Publication year	−0.0001	−0.0004	0.0003	−0.39	0.6959
		Sample size	0.0004	−0.0011	0.0018	0.50	0.6185
** *TaqI* **	t vs. T	Publication year	− 0.0000	−0.0003	0.0002	−0.31	0.7589
		Sample size	−0.0004	−0.0016	0.0009	−0.57	0.5696
	tt vs. TT	Publication year	<0.0001	−0.0003	0.0004	0.22	0.8294
		Sample size	−0.0015	−0.0030	0.0001	−1.86	0.0630
	Tt vs. TT	Publication year	−0.0001	−0.0006	0.0005	−0.22	0.8247
		Sample size	−0.0002	−0.0028	0.0024	−0.13	0.8970
	tt + Tt vs. TT	Publication year	−0.0001	−0.0005	0.0004	−0.23	0.8177
		Sample size	−0.0003	−0.0027	0.0020	−0.29	0.7707
	tt vs. TT + Tt	Publication year	− 0.0000	−0.0004	0.0003	−0.13	0.9005
		Sample size	−0.0010	−0.0025	0.0004	−1.40	0.1611

CI: confidence interval.

## Data Availability

The datasets used and/or analyzed during the current study are available from the corresponding author on reasonable request.

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
