# Peer review of "Association of VDR Polymorphisms (FokI, ApaI, and TaqI) with Susceptibility to Lumbar Disc Herniation: Systematic Review, Meta-Analysis, Trial Sequential Analysis, and Transcriptional Prediction"

_medicina, 2025, doi:10.3390/medicina61050882_

Round 1
Reviewer 1 Report
Comments and Suggestions for Authors
The article, Identifies TaqI as a potential biomarker for LDH risk, suggesting avenues for personalized medicine. Subgroup analyses highlight the protective role of TaqI in Asian populations, adding nuance to genetic associations. Here are my comments,
- Significant heterogeneity (I²up to 83%) in FokI analyses undermines confidence in pooled estimates.
- Some studies (e.g., Gaydarski 2023 for FokI) deviate from HWE, suggesting genotyping errors or population stratification.
- TSA reveals that sample sizes for FokIand some TaqI models fall below the required information size (RIS).
- FATHMM-XF predicts all variants (FokI, ApaI, TaqI) as "benign," contradicting the meta-analysis results for TaqI.
- No meta-regression for confounders (e.g., vitamin D levels, BMI, occupational factors).
Author Response
Dear Reviewer #1, thank you for all your kind efforts. Please find attached the detailed point-by-point-response.
Sincerely,
Serge Brand

Reviewer 2 Report
Comments and Suggestions for Authors
- In the introduction I would add the percentage of cauda equina symptoms
- - In the results Figure 1 19 records were excluded: can you explain the reason?
Author Response
Dear Reviewer #2; thank you for all your kind efforts. Please find attached the detailed point-by-point response.
Sincerely,
Serge Brand

Round 2
Reviewer 1 Report
Comments and Suggestions for Authors
The revised version is suitable for publication.